# Playing in the UEFA Europa League Does Not Adversely Affect English Premier League or La Liga Performance

**DOI:** 10.3390/jfmk4010002

**Published:** 2018-12-22

**Authors:** Richard Stephens, Grant Bosworth, Thomas A. Shepherd

**Affiliations:** 1Centre for Psychological Research, Keele University, Staffordshire ST5 5BG, UK; 2Research Institute for Primary Care and Health Sciences, Keele University, Staffordshire ST5 5BG, UK

**Keywords:** football, soccer, confirmatory bias, UEFA Cup, midweek matches

## Abstract

Purpose: This article presents data challenging the widely held but untested view that concurrently playing mid-week Union of European Football Associations (UEFA) Europa League matches adversely affects domestic league performance. Method: Over 16 seasons we compared next-season domestic league performance of the two highest finishing UEFA Europa League qualifying clubs with the two highest finishing non-qualifiers in England and Spain. Results: Clubs concurrently playing UEFA Europa League football showed significantly superior domestic league performance including wins, losses, goals, goal difference and points tally. The number of European matches played was not related to domestic league performance. Conclusions: The absence of prior rigorous analysis including appropriate comparison data has led to the proliferation of a widespread confirmatory bias, defined as the tendency to seek out evidence that backs one’s hunches and to ignore evidence that contradicts them. Based on our evidence, football professionals, journalists, pundits, and fans should consider the UEFA Europa League competition more favorably.

## 1. Introduction

A compelling narrative has arisen in contemporary football (soccer) commentary around the alleged negative impacts on national domestic league performance of concurrently contesting the Union of European Football Associations (UEFA) Europa League competition [1]. Based on virtually no systematic analysis of data, the Europa League has been described as a ‘poisoned chalice’, a ‘secondary club competition’ and an ‘unwanted distraction’ from league football [2]. This is due to factors including additional fixtures in an already busy schedule, Saturday games being moved to Sundays to accommodate UEFA fixtures played on Thursday evenings, as well as increased risk of injuries and accumulated player fatigue [3,4].

Academic research has contributed towards the negative impact narrative. Cox, Gilmore and Graham report on a sample of four English Premier League (EPL) clubs that had played in the UEFA Europa League, a competition previously known as the UEFA Cup and henceforth referred to in this article as the UEFA EL/C, for two or more seasons from 2005-6 to 2012-3 [5]. They observed that three of these clubs finished in lower domestic league positions in seasons playing both competitions concurrently, compared with seasons when not. Furthermore, Verheijen, a football consultant, reports an analysis of 27,000 top flight games across seven countries and ten years, finding that clubs contesting the UEFA EL/C recoup an average of 1.27 points in domestic league matches played on Sundays after playing UEFA EL/C games on Thursdays, a lower average than the 1.68 points recouped, on average, for all domestic league games [6]. This indicates a deficit of 0.41 points per domestic league game following UEFA EL/C fixtures. 

While apparently supporting the negative impact narrative, both studies are limited, the former by a very small and unrepresentative sample and the latter by the absence of basic descriptive data such that means were presented in the absence of variance estimates. A further issue with both studies is the absence of an appropriate comparison or control condition. A fairer assessment of the impact of concurrent UEFA EL/C football on domestic league performance should compare clubs that are in the UEFA EL/C with clubs at a comparable domestic level that are not so engaged. As no such analysis has yet been undertaken, we remain skeptical that concurrently playing in the UEFA EL/C impacts negatively on domestic league form. Our skepticism was also aroused by data from prior studies showing that fixture congestion, defined as six matches in 21 days or three matches in four days, does not affect the physical and technical aspects of player performance [7]. Such congestion is associated with a higher rate of injuries [7] but, arguably, negative effects of injuries may be managed by ensuring adequate cover in the first team squad. 

We conducted the present study with the aim of providing a more critical analysis. Just as previous research has analyzed real world golf performance on the Professional Golfers’ Association tour [8] we found it relatively straightforward to identify and analyze online sources of data pertinent to domestic league performance in football. We included both the English Premier League (EPL) and the Spanish La Liga as each contains 20-member clubs and because these countries are ranked first and second with respect to performance in UEFA competitions [9]. However, we note the differences in the style of football played in these respective leagues, with the EPL characterized as having a “direct style” and La Liga characterized as being more “aesthetic” (with more emphasis on entertainment than results) combined with demanding “strong ball control” [10].

## 2. Materials and Methods 

In England and Spain, between two and four clubs qualify for the UEFA EL/C each season. We compared EPL and La Liga performance of the two clubs placed highest the previous season that were concurrently playing UEFA EL/C football with that of the two highest-placed clubs that avoided UEFA EL/C and UEFA Champions League qualification. Usually the fifth and sixth placed clubs automatically qualify while the seventh placed club is able to qualify provided it wins its games in one or more preliminary rounds. This means that usually the two highest qualifiers finish fifth and sixth and the two comparison non-qualifiers finish seventh and eighth, or if the seventh placed club qualified through the preliminary rounds, eighth and ninth. However, some seasons deviate from this pattern. In 2003-04 Middlesbrough were the second qualifiers from eleventh place as League Cup winners. In 2004-05 the first and second qualifiers finished sixth and seventh because Liverpool qualified for the UEFA Champions League from fifth as winners the previous season. In 2010-11 Fulham were the second highest qualifiers from eighth position based on their UEFA Respect Fair Play ranking. In 2012-13 Swansea were the second highest placed UEFA EL/C qualifiers in ninth position as League Cup winners. In 2016-17 Arsenal were the second highest qualifiers from seventh position because Manchester United qualified for the UEFA Champions League from sixth as UEFA Europa League winners. Our analysis extended back over the last 16 seasons to reflect the modern game as much as possible while still providing sufficient data for our analyses to be meaningful.

Data pertaining to league performance were obtained from the official websites of the Premier League [11] and La Liga [12]. For the full data set please refer to “Appendix A” in the Appendix A. Data on managerial appointments, as well as UEFA EL/C qualification and games played were obtained from the worldfootball.net website [13]. This website is run by HEIM:SPIEL Medien GmbH & Co, a commercial company specializing in online sports data with customers including Sky Sports, FC Barcelona, and the Bundesliga [14]. As this study analyses data that is freely available and in the public domain, namely English and Spanish elite football league table data, formal Research Ethics Committee approval was not required.

The main analyses consisted of a series of independent samples *t*-tests comparing, across the seasons 2002-03 to 2017-18, the two-highest placed UEFA EL/C qualifying clubs per season from the EPL (*n* = 32) versus the two-highest placed EPL clubs that avoided UEFA EL/C qualification (*n* = 32). The key dependent variables were wins, draws, losses, goals for, goals against, points tally and change in league position for domestic league football. We also examined two variables pertinent to in-season performance. The first was the number of changes of manager mid-season, between the dates of 1st July, to include pre-season, and mid-May. This was on the assumption that changing manager is an indication of response to crisis, and therefore that a higher number of managers is a negative outcome. The second was the total number of UEFA EL/C games played, on the assumption that playing a higher number of European games might disrupt domestic league performance. All analyses were subsequently repeated including the equivalent data from La Liga.

## 3. Results

### 3.1. Domestic League Performance

Descriptive data are provided in Table 1 and Table 2. Within the EPL there was one significant difference such that clubs concurrently playing UEFA EL/C and domestic football scored more domestic league goals than non-UEFA EL/C qualifiers, *t* (62) = 2.226, *p* = 0.030, *d* = 0.565, 95% *CI* = 0.614, 11.448. Combining the EPL and La Liga data we found that the UEFA EL/C qualifiers won more domestic league games, *t* (126) = 3.108, *p* = 0.002, *d* = 0.554, 95% *CI* = 0.789, 3.555, lost fewer games, *t* (126) = 2.240, *p* = 0.027, *d* = 0.399, 95% *CI* = −2.943, −0.192, scored more goals, *t* (126) = 3.854, *p* < 0.001, *d* = 0.687, 95% *CI* = 3.368, 10.476, had superior goal difference, *t* (126) = 3.167, *p* = 0.002, *d* = 0.564, 95% *CI* = 3.446, 14.929 and accrued more points, *t* (126) = 2.964, *p* = 0.004, *d* = 0.528, 95% *CI* = 1.963, 9.849, than non-UEFA EL/C qualifiers.

### 3.2. Changes of Manager

For the EPL, a change of manager mid-season was observed in 8 (25.0%) clubs concurrently playing UEFA EL/C and domestic league football and 7 (21.9%) clubs not involved in the UEFA EL/C competitions. These proportions did not differ significantly, chi-square = 0.087, *df* = 1, *n* = 64, *p* = 0.768, *w* = 0.037. Adding the La Liga data, these figures were, respectively, 16 (25.0%) and 13 (20.3%). These proportions still did not differ significantly, chi-square = 0.401, *df* = 1, *n* = 128, *p* = 0.526, *w* = 0.056.

### 3.3. Number of UEFA EL/C Matches Played

Finally, the regression of the number of UEFA EL/C games played onto the variable ‘final domestic league points tally the season after qualifying’ in the 32 EPL clubs in our sample that were playing European football showed that there was no significant relationship between these variables, *r* = 0.099, *F* (1, 30) < 1.0, partial eta squared = 0.002. Adding data from the 32 La Liga clubs that were playing European football for the 15 seasons under analysis, there was still no significant relationship, *r* = 0.376, *F* (1, 62) = 1.089, Mean Square Error = 131.747, *p* = 0.301, partial eta squared = 0.017.

## 4. Discussion

In this paper we have presented analyses of domestic league performance of football clubs in England’s top division, the EPL, and Spain’s top division, La Liga. We compared clubs that qualified for the UEFA Europa League or Cup and consequently played in these competitions alongside performing in their domestic leagues with clubs that narrowly avoided UEFA EL/C qualification and so were not concurrently playing European and domestic league fixtures. For both the English and Spanish clubs assessed there was no evidence that playing in the UEFA Cup or UEFA Europa League conferred any disadvantage in terms of concurrent domestic league performance, but rather, the opposite. 

Our data indicate some advantage of qualifying highly enough in the league to be eligible for the UEFA EL/C that carried over to the following season. This included scoring an average of five more goals across the season for EPL clubs, and for the dataset that included the EPL and La Liga, winning two more games, losing between one and two fewer games, scoring six more goals and accruing between five and six more points. We were not altogether surprised by this finding because, as noted in the Introduction, one of the commonly cited negative aspects of contesting the UEFA EL/C, fixture congestion, has been shown in previous research not to affect physical and technical aspects of player performance [7].

Considering possible limitations of our study, one issue inherent in our design is that the EPL and La Liga clubs in our analyses that were concurrently playing domestic league and UEFA EL/C football finished the prior season an average of 2.28 places (*SD* = 1.1) above the comparator clubs that narrowly missed out on UEFA EL/C football. It might be questioned whether the superior performance observed is still below what might be achieved in the absence of Thursday evening UEFA EL/C football. It was suggested during peer review that we might compare next season performance of clubs finishing fifth and sixth with those finishing seventh and eight for comparable professional leagues where these positions do not confer UEFA EL/C qualification. After careful consideration we selected the second tier of professional football in England, known as the Championship, for this purpose. The Championship meets the criterion of not conferring UEFA EL/C qualification, while nevertheless being financially comparable with the EPL and La Liga. The Championship has the third largest number of live spectators (11.1 M) across all European leagues, above La Liga (10.6 M) and behind only Germany’s Bundesliga (12.7 M) and the EPL (13.6 M) [15]. The Championship is also the sixth highest paying league in Europe based on the amount clubs spend on player wages [15]. Added to this the number of clubs that contest the Championship, at 24, is also comparable to the EPL and La Liga. 

Pertinent data were obtained from the worldfootball.net website [13]. For the full data set please refer to “Appendix A” in the Appendix A. In 8 of the 16 seasons included in these analyses, the fifth or sixth placed Championship club won promotion to the EPL in end of season play-offs and so was absent from the league the following season. To maintain a consistent sample size, we replaced the promoted fifth or sixth placed club with the club that finished fourth the previous season. Comparing Championship clubs that finished the preceding season in fourth, fifth and/or sixth positions with those placed seventh and eighth, there was a significant difference in final league position, *t* (62) = 2.061, *p* = 0.043, *d* = 0.523, 95% *CI* = -6.586, -0.101. This was such that the lower placed clubs gained an average of 3.34 (SD = 9.17) additional league places the following season. There were no differences for wins, draws, losses, goals for, goals against, goal difference and points tally, *t* (62) < 0.696, *p* > 0.489, *d* < 0.18. 

These additional analyses contextualize our results by demonstrating that league position the previous season is not a reliable predictor of league performance the following season. On this basis, league position the previous season does appear to be the best explanation for the superior domestic league performance of the EPL and La Liga clubs concurrently contesting the UEFA EL/C, compared with clubs that just missed out on qualification. Presumably, the superior domestic league performance may be a combination of fitness afforded by the additional game time and/or strengthening of squads in preparation for an increased number of games. Further analysis, beyond the scope of this paper, would be required to investigate this in more detail.

There were no differences in change of manager, a variable that might be taken to indicate response to crisis, although the trend was for more changes of manager in UEFA EL/C qualifying clubs. This may reflect greater pressure on clubs concurrently contesting the UEFA EL/C due to the perception of a negative impact of domestic league form. Additionally, the number of UEFA EL/C matches played indicates how far in the competition a club has progressed and provides an indication of how seriously a club was contesting this competition. This is an important consideration given that some clubs notoriously field weakened teams for UEFA EL/C matches [16]. However, for the clubs concurrently playing domestic and UEFA EL/C football, there was no relationship between the number of UEFA EL/C matches played and final domestic league points tally, providing no support for the notion that contesting the UEFA EL/C competitions detracts from domestic league performance. 

The data from La Liga clarify and extend the pattern of findings observed in the EPL, most notably by broadening the number of variables identified by inferential statistical analysis as showing superior domestic league performance among the UEFA EL/C qualifying clubs. This finding indicates commonality in performance outcomes of the EPL and La Liga despite the acknowledged more direct style of football that characterizes the former and the more aesthetic style that characterizes the latter [10]. We add no further comment with respect to La Liga itself, except to reiterate that the burden of additional fixtures and associated risk of injury and fatigue associated with UEFA EL/C qualification are similarly pertinent in Spain as they are in England.

Our paper highlights a notable example in professional football commentary of what we interpret to be confirmatory bias, which is a tendency to seek out evidence that backs hunches and to ignore evidence that contradicts them [17]. Specifically, this relates to an unsubstantiated claim, contradicted by our analysis, that concurrently playing in the UEFA Europa League is detrimental to domestic league performance relative to clubs at a similar level with no such commitment. We have observed examples of this in reputable sources with large readerships including the British Broadcasting Company (BBC) [4], the Guardian [18], the Independent [19], the Huffington Post [20], and the Daily Mail [21]. 

Under confirmatory bias, commentators highlight exemplars that fuel the negative impact narrative and ignore contradictory cases. So, for example, it is pointed out that ‘Liverpool excelled domestically when void of Europa distraction’ or that ‘Manchester United’s league form has improved without any European action’ [18]. However, these changes in fortune can just as plausibly be accounted for by alternative explanations. Liverpool in 2013-14 benefitted greatly from the exceptional form of the Uruguayan forward Luis Suárez [22], while Manchester United in 2014-15 were recovering from an abject period under the management of David Moyes the ill-fated replacement for hugely successful manager Sir Alex Ferguson [23]. Meanwhile, it goes relatively ignored when clubs that narrowly miss out on UEFA EL/C qualification are not able to out-compete clubs with European commitments the following season, but instead play so poorly as to finish in one of the relegation places. This happened to Real Club Deportivo Mallorca in 2011-12 and Birmingham City in 2009-10 [13]. It appears that poor domestic results for clubs concurrently contesting the UEFA EL/C and upsurges in domestic performance of clubs that avoid UEFA EL/C qualification most likely reflect the natural ebb and flow of form rather than being a direct consequence of European football.

Football is a far-reaching pastime with the power to influence many people, from the millions of fans and amateur players to individuals that make their living from the game. This paper has extended academic understanding of the impact of concurrently playing in the UEFA EL/C competition upon domestic league performance across two of the Europe’s most prestigious domestic league competitions. Our findings indicate that previous studies [5,6] may have over-estimated negative impacts of elite level European football on domestic league performance by not including appropriate control/comparison data. A recent economic analysis has found that clubs can benefit financially from playing in the UEFA Europa League [5]. These benefits seem to go largely unrecognized due to the negative impact “curse of UEFA qualification” narrative that still dominates in football commentary. We hope that the analyses presented in this paper may go some way towards changing this.

## 5. Conclusions

In an analysis of EPL and La Liga final league table data from 2002-03 to 2017-18 we found superior domestic league performance in clubs concurrently playing in the UEFA EL/C competitions, in comparison with clubs that narrowly missed out on UEFA EL/C qualification. Contrary to popular opinion, there was no evidence of poorer domestic league performance due to mid-week European football. We argue that the absence of prior rigorous analysis including appropriate comparison data has led to the proliferation of a widespread confirmatory bias, defined as the tendency to seek out evidence that backs one’s hunches and to ignore evidence that contradicts them. These findings are relevant to a broad range of stakeholders in football including professional players and coaches, individuals involved in the amateur game, as well as journalists, pundits, and fans. Based on the evidence presented we argue that the UEFA Europa League competition should be considered more favorably than hitherto has been the case.

## Figures and Tables

**Table 1 jfmk-04-00002-t001:** Clubs included in the analysis. Numbers in parentheses are the finishing league positions in, first, the preceding (qualifying) season, and second, in the season indicated in column 1.

Season	UEFA Europa League/Cup Qualification	EPL	La Liga
**2002-03**	Qualified#1	Leeds United (5, 15) ^1^	Celta de Vigo (5, 4) ^1^
	Qualified#2	Chelsea (6, 4) ^1^	Betis (6, 8)
	Avoided#1	West Ham United (7, 18)	Sevilla (8, 10)
	Avoided#2	Aston Villa (8, 16) ^1^	Athletic Bilbao (10, 7)
**2003-04**	Qualified#1	Liverpool (5, 4) ^2^	Valencia (5, 1) ^2^
	Qualified#2	Blackburn Rovers (6, 15) ^1^	Barcelona (6, 2) ^2^
	Avoided#1	Everton (7, 17)	Athletic Bilbao (7, 5)
	Avoided#2	Tottenham Hotspur (10, 14)	Betis (8, 9) ^1^
**2004-05**	Qualified#1	Newcastle United (5, 14) ^1^	Athletic Bilbao (5, 8)
	Qualified#2	Middlesbrough (11, 7)	Sevilla (6, 6)
	Avoided#1	Aston Villa (6, 10)	Atlético Madrid (7, 11)
	Avoided#2	Charlton Athletic (7, 11)	Betis (9, 4)
**2005-06**	Qualified#1	Bolton Wanderers (6, 8)	Espanyol (5, 15)
	Qualified#2	Middlesbrough (7, 14) ^1^	Sevilla (6, 5) ^1^
	Avoided#1	Manchester City (8, 15)	Valencia (7, 3) ^1^
	Avoided#2	Tottenham Hotspur (9, 5)	Deportivo La Coruña (9, 8) ^2^
**2006-07**	Qualified#1	Tottenham Hotspur (5, 5)	Sevilla (5, 3) ^1^
	Qualified#2	Blackburn Rovers (6, 10)	Celta de Vigo (6, 18)
	Avoided#1	Bolton Wanderers (8, 7) ^1^	Villarreal (7, 5) ^2^
	Avoided#2	Wigan Athletic (10, 17)	Deportivo La Coruña (8, 13)
**2007-08**	Qualified#1	Tottenham Hotspur (5, 11) ^1^	Villarreal (5, 2)
	Qualified#2	Everton (6, 5)	Zaragoza (6, 18)
	Avoided#1	Reading (8, 18)	Recreativo (8, 16)
	Avoided#2	Portsmouth (9, 8)	Racing Santander (10, 6)
**2008-09**	Qualified#1	Everton (5, 5) ^1^	Sevilla (5, 3) ^2^
	Qualified#2	Aston Villa (6, 6)	Racing Santander (6, 11)
	Avoided#1	Blackburn Rovers (7, 15) ^1^	Mallorca (7, 9)
	Avoided#2	West Ham United (10, 9)	Almería (9, 12)
**2009-10**	Qualified#1	Everton (5, 8) ^1^	Villarreal (5, 7) ^2^
	Qualified#2	Aston Villa (6, 6) ^1^	Valencia (6, 3) ^1^
	Avoided#1	Tottenham Hotspur (8, 14) ^1^	Deportivo La Coruña (7, 10) ^1^
	Avoided#2	West Ham United (9, 17)	Málaga (8, 17)
**2010-11**	Qualified#1	Manchester City (5, 3)	Getafe (6, 16)
	Qualified#2	Aston Villa (6, 9) ^1^	Villarreal (7, 4) ^1^
	Avoided#1	Everton (8, 7) ^1^	Mallorca (5, 17)
	Avoided#2	Birmingham City (9, 18)	Deportivo La Coruña (10, 18)
**2011-12**	Qualified#1	Tottenham Hotspur (5, 4) ^2^	Sevilla (5, 9) ^1^
	Qualified#2	Fulham (8, 9) ^1^	Athletic Bilbao (6, 10)
	Avoided#1	Liverpool (6, 8) ^1^	Espanyol (8, 14)
	Avoided#2	Everton (7, 7)	Osasuna (9, 7)
**2012-13**	Qualified#1	Tottenham Hotspur (4, 5) ^1^	Atlético Madrid (5, 3) ^1^
	Qualified#2	Newcastle United (5, 16)	Levante (6, 11)
	Avoided#1	Everton (7, 6)	Osasuna (7, 16)
	Avoided#2	Fulham (9, 12) ^1^	Mallorca (8, 18)
**2013-14**	Qualified#1	Tottenham Hotspur (5, 6) ^1^	Valencia (5, 8) ^2^
	Qualified#2	Swansea City (9, 12)	Betis (7, 20)
	Avoided#1	Everton (6, 5)	Málaga (6, 11) ^2^
	Avoided#2	Liverpool (7, 2)^1^	Rayo Vallecano (8, 12)
**2014-15**	Qualified#1	Everton (5, 11)	Sevilla (5, 5) ^1^
	Qualified#2	Tottenham Hotspur (6, 5) ^1^	Villarreal (6, 6)
	Avoided#1	Manchester United (7, 4) ^2^	Valencia (8, 4) ^1^
	Avoided#2	Southampton (8, 7)	Celta Vigo (9, 8)
**2015-16**	Qualified#1	Tottenham Hotspur (5, 3) ^1^	Villarreal (6, 4) ^1^
	Qualified#2	Liverpool (6, 8) ^2^	Athletic Bilbao (7, 5) ^1^
	Avoided#1	Swansea City (8, 12)	Celta Vigo (8, 6)
	Avoided#2	Stoke City (9, 9)	Málaga (9, 8)
**2016-17**	Qualified#1	Manchester United (5, 6) ^2^	Athletic Bilboa (5, 7) ^1^
	Qualified#2	Southampton (6, 8) ^1^	Celta Vigo (6, 13)
	Avoided#1	Liverpool (8, 4) ^1^	Malaga (8, 11)
	Avoided#2	Stoke City (9, 13)	Real Sociedad (9, 6)
**2017-18**	Qualified#1	Arsenal (5, 6) ^2^	Villarreal (5, 5) ^1^
	Qualified#2	Everton (7, 8)	Real Sociedad (6, 12)
	Avoided#1	Southampton (8, 17) ^1^	Espanyol (8, 11)
	Avoided#2	Bournemouth (9, 12)	CD Alavés (9, 14)

^1^ Played UEFA Europa League/Cup football in season of qualification. ^2^ Played UEFA Champions League football in season of qualification. Abbreviations: UEFA = Union of European Football Associations; EPL = English Premier League.

**Table 2 jfmk-04-00002-t002:** Domestic league performance for clubs from the EPL, La Liga, and both leagues combined.

	Domestic Competitions and UEFA EL/C Played Concurrently			Domestic Competitions Only		
	EPL	La Liga	EPL/La Liga	EPL	La Liga	EPL/La Liga
**Number of clubs**	32	32	64	32	32	64
**Games won**	15.63	16.22	15.92	13.84	13.66	13.75 **
	(3.38)	(4.63)	(4.03)	(4.55)	(3.13)	(3.88)
**Games drawn**	10.09	8.75	9.42	9.94	10.13	10.03
	(2.55)	(1.97)	(2.36)	(2.77)	(2.88)	(2.81)
**Games lost**	12.28	13.03	12.66	14.22	14.22	14.22 *
	(3.75)	(4.07)	(3.90)	(4.29)	(3.74)	(3.99)
**Goals for**	55.41	53.34	54.38	49.38 *	45.53	47.45 **
	(7.98)	(9.30)	(8.66)	(13.09)	(9.39)	(11.46)
**Goals against**	48.28	47.78	48.03	50.28	50.00	50.14
	(9.59)	(11.42)	(10.46)	(9.26)	(10.92)	(10.04)
**Goal difference**	7.13	5.56	6.34	−0.91	−4.78	−2.84 **
	(14.29)	(17.65)	(15.95)	(19.06)	(14.38)	(16.86)
**Points tally**	56.97	57.41	57.19	51.47	51.09	51.28 **
	(9.88)	(13.05)	(11.49)	(12.81)	(9.17)	(11.05)
**Change in League Position**	−2.16	−2.22	−2.19	−2.75	−2.13	−2.44
	(3.80)	(5.01)	(4.41)	(4.83)	(4.71)	(4.74)
**UEFA EL/C games played**	9.44	10.28	9.86	-	-	-
	(4.01)	(4.04)	(4.02)			

* Significant difference *p* < 0.05. ** Significant difference *p* < 0.01. Analyses are for the comparison of clubs concurrently playing domestic league and UEFA EL/C vs. clubs playing in domestic league only; these were assessed for EPL only, and for EPL and La Liga combined.

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
