# Peer review of "Playing in the UEFA Europa League Does Not Adversely Affect English Premier League or La Liga Performance"

_jfmk, 2018, doi:10.3390/jfmk4010002_

Reviewer 1 Report

Having read the article "Playing in the UEFA Europa League does not adversely affect English Premier League or La Liga performance", I believe there can be a good contribution for the current state of the art. Nerveless, there are some aspects that need a carefully attention by the authors, this is the reason why I recommend revisions at this stage:

1. The abstract should be reformulated. There are no objective information concerning the purpose of the article, and details about the methodological aspects

2. The keywords should be complementary from the title and not repeated.

3. The introduction section should be improved. The authors presented a general view of the state of the art. Some important and major references are missed. For example, see the studies of Sarmento et al., for a current state of the art, and for an explanation about differences in the study of play in each one of the studied leagues that are important to understand this specific topic.

Sarmento, H., Clemente, F. M., Araújo, D., Davids, K., McRobert, A., & Figueiredo, A. (2018). What Performance Analysts Need to Know About Research Trends in Association Football (2012–2016): A Systematic Review. Sports Medicine, 48(4), 799-836. doi:10.1007/s40279-017-0836-6

Sarmento, H., Pereira, N., Matos, N., Campaniço, N., Anguera, M., & Leitão, J. (2013). English Premier League, Spain´s La Liga and Italy´s Serie´s A – What´s Different? International Journal of Performance Analysis in Sport, 13, 773-789.

4. The last paragraph of introduction should be placed in methods section.

5. Results – Good presentation. Well done!

6. Discussion – A more deeply discussion is required. The authors can use the above-mentioned references to do it.

7.  Conclusion - “Based on our evidence, football professionals, journalists, pundits and fans might come to view the UEFA Europa League competition more favorably.” This a conclusion of a scientific study? Are you working to change the views of journalists and so one? 

Author Response

Having read the article "Playing in the UEFA Europa League does not adversely affect English Premier League or La Liga performance", I believe there can be a good contribution for the current state of the art. Nerveless, there are some aspects that need a carefully attention by the authors, this is the reason why I recommend revisions at this stage:

1. The abstract should be reformulated. There are no objective information concerning the purpose of the article, and details about the methodological aspects

We have updated the Abstract to explicitly state the purpose of the article (please see first line) and we provide more methodological details (please see lines 3-5).

2. The keywords should be complementary from the title and not repeated. 

            We have updated the keywords to avoid repetition of words included in the title.

3. The introduction section should be improved. The authors presented a general view of the state of the art. Some important and major references are missed. For example, see the studies of Sarmento et al., for a current state of the art, and for an explanation about differences in the study of play in each one of the studied leagues that are important to understand this specific topic.

Sarmento, H., Clemente, F. M., Araújo, D., Davids, K., McRobert, A., & Figueiredo, A. (2018). What Performance Analysts Need to Know About Research Trends in Association Football (2012–2016): A Systematic Review. Sports Medicine, 48(4), 799-836. doi:10.1007/s40279-017-0836-6

Sarmento, H., Pereira, N., Matos, N., Campaniço, N., Anguera, M., & Leitão, J. (2013). English Premier League, Spain´s La Liga and Italy´s Serie´s A – What´s Different? International Journal of Performance Analysis in Sport, 13, 773-789. 

Thank you for alerting us to these useful papers. We now reference each in the Intro and Discussion. Full citations are added to the References section.

4. The last paragraph of introduction should be placed in methods section.

We have moved the final paragraph of the Introduction to the Method section as requested.

5. Results – Good presentation. Well done!

Many thanks.

6. Discussion – A more deeply discussion is required. The authors can use the above-mentioned references to do it.

Thank you. We have expanded the Discussion to include the above papers and also in response to some of reviewer 2’s suggestions.

7.  Conclusion - “Based on our evidence, football professionals, journalists, pundits and fans might come to view the UEFA Europa League competition more favorably.” This a conclusion of a scientific study? Are you working to change the views of journalists and so one? 

We wish to disseminate the findings of our study to as wide an audience as possible including stakeholders such as football professionals, journalists, pundits and fans. We would also wish to impress on people that might not usually engage with science that taking an evidence-based approach can lead to better decision making in all spheres of life – even football!. Still, upon considering this sentence in light of your comment, we have revised the wording (please see page 7, lines 247-250).

Reviewer 2 Report

This analysis attempts to measure whether playing in the Europa League negatively affects domestic league play.  The question is valid and interesting.  The analysis ends up being a half-measure in that a team in Europa League may in fact play worse in its domestic league given the findings, but not so poorly that it plays worse than the teams just below it in the table/rankings.

1.      Why is Champions League not tested too?  Please explain.

2.      With respect to choosing similar clubs that did not qualify, please explain how the clubs are chosen by UEFA, as I note in 2002-03 the 5th and 6th EPL finishers qualified and 7th and 8th did not.  That seems as expected, yet in the subsequent season, it was 5th and 6th qualifying, but then 7th and 10th not qualifying.  Where is 8th and 9th?  Please explain.

3.      Similarly, if the better ranked teams tend to qualify more than the worse ranked teams, how is that really a comparison of similar clubs?  Finishing 5th is much different than finishing 8th.  It might be that the 5th/6th place team doesn’t perform worse than the 7th/8th place team simply because the former do perform a bit worse and slide down to the level of the 7th/8th place teams.

a.      An alternative is to see if you can compare 5th/6th place teams to 7th/8th place teams where none of the teams made Europa League (or do those teams not exist).

b.      For instance, is there a 3rd league (e.g., Danish league or Norwegian league) where you could compare 5th/6th to 7th/8th because maybe that league only sends teams in top 4 to Champions League and Europa League.  Then you could see if 5/6 perform better than 7/8 in general.  If yes, then your current analysis would indicate that the 5/6 team performs worse than expected.  If not, then the opposite.

4.      It’s a half measure…in other words, if the research did find that UEFA League clubs performed worse, then that would be consistent with the pundits.  If they don’t perform worse (as is shown), then that might be because the comparator clubs are not as good since they finished worse anyway.

5.      An alternate model is run a regression of Points = f(Points(t-1), UEFA League dummy variable).  This allows for the same team in the previous season to be used as a comparable as well as other teams.  Dickey-Fuller type tests, etc., would need to be done too.

Author Response

Reviewer 2

Are the   conclusions supported by the results?

( )

(x)

( )

( )

Comments and Suggestions for Authors

This analysis attempts to measure whether playing in the Europa League negatively affects domestic league play.  The question is valid and interesting.  The analysis ends up being a half-measure in that a team in Europa League may in fact play worse in its domestic league given the findings, but not so poorly that it plays worse than the teams just below it in the table/rankings.

1.      Why is Champions League not tested too?  Please explain.

Our paper breaks from the approach used previously of comparing a club’s domestic league position one season when they were concurrently contesting the UEFA Europa League with its position the prior season when it was not in Europe. This is on the basis that this method does not take account of year to year fluctuations. If we had only compared the EPL and La Liga clubs concurrently contesting the UEFA Europa League with their domestic performance the previous season, then we would have reported that they dropped an average of 2.2 places and attributed all of this to the “burden” of concurrently contesting domestic and European competition. However, our inclusion of comparison clubs that did not qualify for the UEFA Europa League provides a context against which to better appreciate the implication of this deficit, when we see that these comparison clubs also fell by about the same amount, 2.4 places.

Our novel method of assessing the impact of concurrently contesting the UEFA Europa League on domestic league performance relies on a tier of clubs just below the qualifying clubs that do not have a European commitment to act as comparators. This would not be possible for UEFA Champions League clubs as their immediate competitors contest the UEFA Europa League. That is why this paper is restricted to the UEFA Europa League and does not include the UEFA Champions League.

2.      With respect to choosing similar clubs that did not qualify, please explain how the clubs are chosen by UEFA, as I note in 2002-03 the 5th and 6th EPL finishers qualified and 7th and 8th did not.  That seems as expected, yet in the subsequent season, it was 5th and 6th qualifying, but then 7th and 10th not qualifying.  Where is 8th and 9th?  Please explain.

We have added some text explaining the complexities of UEFA Europa League qualification on page 2, lines 69-81 (Materials and Methods).

3.      Similarly, if the better ranked teams tend to qualify more than the worse ranked teams, how is that really a comparison of similar clubs?  Finishing 5th is much different than finishing 8th.  It might be that the 5th/6th place team doesn’t perform worse than the 7th/8th place team simply because the former do perform a bit worse and slide down to the level of the 7th/8th place teams.

a.      An alternative is to see if you can compare 5th/6th place teams to 7th/8th place teams where none of the teams made Europa League (or do those teams not exist).

b.      For instance, is there a 3rd league (e.g., Danish league or Norwegian league) where you could compare 5th/6th to 7th/8th because maybe that league only sends teams in top 4 to Champions League and Europa League.  Then you could see if 5/6 perform better than 7/8 in general.  If yes, then your current analysis would indicate that the 5/6 team performs worse than expected.  If not, then the opposite.

Thanks – we liked the idea of running an analysis for one or more leagues where 5th place onwards are not involved in UEFA EL/C football as a way of further contextualising the effects that we found. However, some thought was required around which leagues. We looked at the top 12 leagues in the UEFA country rankings (https://www.uefa.com/memberassociations/uefarankings/country/#/yr/2019) but all of these had clubs finishing 5th or lower competing in the UEFA EL/C. The same was the case for the 13th country which was the first you suggest, Denmark, with respect to 5th position. In addition, the Danish Superliga comprised just 12 clubs until it was expanded to 14 in 2016-17. In this league clubs finishing 5-8th are mid-table rather than towards the top of the league table, as is the case for the EPL and La Liga, and as such, may perform differently. We also decided against running analyses for the 16-club Norwegian Eliteserien because of the financial gulf between this league and the EPL/ La Liga. For example, the average player wage bill for clubs finishing outside the top four in Norway is 4M Euros per annum compared with a figure of 164M for equivalent clubs from the EPL, and 60m for equivalent clubs from La Liga. After consideration, we identified the English Championship as a more appropriate comparison given its very strong financial status and the number of clubs that contest it.
Please see the Discussion, page 5-6, lines 154-187 for this analysis and its interpretation.

4.      It’s a half measure…in other words, if the research did find that UEFA League clubs performed worse, then that would be consistent with the pundits.  If they don’t perform worse (as is shown), then that might be because the comparator clubs are not as good since they finished worse anyway.

Please see response to point 3.

5.      An alternate model is run a regression of Points = f(Points(t-1), UEFA League dummy variable).  This allows for the same team in the previous season to be used as a comparable as well as other teams.  Dickey-Fuller type tests, etc., would need to be done too.

We do not support this approach from a theoretical perspective based on the analyses presented for point 3 in which we show that league position the previous season is not a reliable predictor of league performance the following season. Moreover, the issue that we refute is not only the claim that UEFA EL/C clubs are negatively affected in their domestic league, but also the claim that clubs just missing out on UEFA EL/C qualification have an advantage over the unfortunates just above them in the previous year’s league table. Our analyses clearly show this not to be the case. We anticipate an element of debate around our study design and the inclusion of data from The Championship but we feel that the data we present go as far as we can in addressing the possible confounding effect of previous season league position that you have identified. Indeed, we welcome such debate and hope that it encourages people to engage with scientific reasoning.

We would also not support this suggested approach pragmatically, because it ignores the eventuality that clubs may qualify for the UEFA EL/C over consecutive seasons, such as was the case for Tottenham Hotspur who contested the UEFA EL/C over the five seasons 2011-12 to 2015-16. Our study design takes account of this because the point of comparison is a club at a similar level not playing UEFA EL/C football in the same season.

Round  2

Reviewer 2 Report

The major issue I had has been resolved through further analysis.  The minor questions/concerns were also addressed.